# Identification of natural antiviral drug candidates against Tilapia Lake Virus: Computational drug design approaches

Md Afsar Ahmed Sumon[1], Amer H. Asseri[2], Mohammad Habibur Rahman Molla[3]*, Mohammed Othman Aljahdali[3], Md. Rifat Hasan[4], M. Aminur Rahman[5], Md. Tawheed Hasan[6], Tofael Ahmed Sumon[7], Mohamed Hosny Gabr[1], Md. Shafiqul Islam[8], Burhan Fakhurji[9], Mohammed Moulay[10], Earl Larson[11], Christopher L. Brown[12]*

1 Department of Marine Biology, Faculty of Marine Sciences, King Abdulaziz University, Jeddah, Saudi Arabia, 2 Department of Biochemistry, Faculty of Sciences, King Abdulaziz University, Jeddah, Saudi Arabia, 3 Department of Biology, King Abdulaziz University, Jeddah, Saudi Arabia, 4 Department of Applied Mathematics, Faculty of Science, Noakhali Science and Technology University, Noakhali, Bangladesh, 5 Department of Fisheries and Marine Bioscience, Faculty of Biological Science and Technology, Jashore University of Science and Technology, Jashore, Bangladesh, 6 Department of Aquaculture, Sylhet Agricultural University, Sylhet, Bangladesh, 7 Department of Fish Health Management, Sylhet Agricultural University, Sylhet, Bangladesh, 8 Institute of Marine Sciences, University of Chittagong, Chittagong, Bangladesh, 9 iGene Medical Training and Molecular Research Center, Jeddah, Saudi Arabia, 10 Embryonic Stem Cell Research Unit, King Fahd Medical Research Center, King Abdulaziz University, Jeddah, Saudi Arabia, 11 Department of Microbiology, St Johns River State College, Orange Park, FL, United States of America, 12 FAO World Fisheries University Pilot Programme, Pukyong National University, Busan, South Korea

* mrahmanmolla@stu.kau.edu.sa (MHRM); brownchristopher38@gmail.com (CLB)

**Data Availability Statement:** All relevant data are within the paper and its Supporting Information files.

## Abstract

Tilapia Lake Virus (TiLV) is a disease that affects tilapia fish, causing a high rate of sudden death at any stage in their life cycle. Unfortunately, there are currently no effective antiviral drugs or vaccines to prevent or control the progression of this disease. Researchers have discovered that the CRM1 protein plays a critical function in the development and spreading of animal viruses. By inhibiting CRM1, the virus's spread in commercial fish farms can be suppressed. With this in mind, this study intended to identify potential antiviral drugs from two different tropical mangrove plants from tropical regions: *Heritiera fomes* and *Ceriops candolleana*. To identify promising compounds that target the CRM1 protein, a computer-aided drug discovery approach is employed containing molecular docking, ADME (absorption, distribution, metabolism and excretion) analysis, toxicity assessment as well as molecular dynamics (MD) simulation. To estimate binding affinities of all phytochemicals, molecular docking is used and the top three candidate compounds with the highest docking scores were selected, which are CID107876 (-8.3 Kcal/mol), CID12795736 (-8.2 Kcal/mol), and CID12303662 (-7.9 Kcal/mol). We also evaluated the ADME and toxicity properties of these compounds. Finally, MD simulation was conducted to analyze the stability of the protein-ligand complex structures and confirm the suitability of these compounds. The computational study demonstrated that the phytochemicals found in *H. fomes* and *C. candolleana* could potentially serve as important inhibitors of TiLV, offering practical utility. However,

**Funding:** This research was sponsored by Institutional Fund Projects under grant no. IFPIP: 874-130-1443, to author Amer H. Asseri. The authors gratefully acknowledge technical and financial support provided by the Ministry of Education and King Abdulaziz University, DSR, Jeddah, Saudi Arabia. These funders reviewed and approved the project concept but the funders had no role in study design, data collection and analysis, decision to publish, or preparation of the manuscript.

**Competing interests:** There are no competing interests.

further *in vivo* investigations are necessary to investigate and potentially confirm the effectiveness of these compounds as antiviral drugs against the virus TiLV.

## Introduction

Tilapia is one of the most widely consumed, rapidly growing, and profitable families in worldwide aquaculture industries. Tilapia generally have desirable attributes including fast growth, tolerance of environmental changes, stress and disease resistance, ease of captive breeding, high protein content, and palatability [1]. However, aquatic viruses greatly impact the commercial aquaculture industry by causing sudden mortality, resulting in serious economic losses. Tilapia Lake Virus (TiLV) disease is one of the emerging and transboundary diseases of tilapia culture, causing up to 90% sudden mortality at any stage of culture [2]. Several tilapias, including the Nile (*Oreochromis niloticus*), Red (*Oreochromis* sp.), and hybrid tilapia derived from *O. niloticus* and *O. aureus* have been subject to infection with the TiLV [3]. With the increase in surveillance and diagnostic techniques in recent years, it is established that TiLV disease has spread across Asia, Africa, and North and South America [4]. TiLV is caused by a single-stranded (ss)-RNA virus, and its infection is characterized by various symptoms including lethargy, loss of appetite, respiratory distress, skin erosion, and brain lesions. Unfortunately, there are currently no effective therapeutic drugs available to control the spread of this virus [5].

The development of drugs that effectively target host proteins essential to the viral life cycle completion is a promising strategy for antiviral therapy development. CRM1 is a nuclear transporter protein that is understood to have a vital function in various virus families such as coronaviruses, retroviruses, flaviviruses, orthomyxoviruses, rhabdoviruses, paramyxoviruses, and herpesviruses [6]. CRM1 is among the most well-characterized Exportin-1 compounds (XPO1); CRM1-mediated export is used by a variety of viral species at various stages of their life cycles to ensure proper protein localization [7]. When CRM1, a protein responsible for cellular transport is blocked, the levels of Protein 4 in the cytoplasm and nuclei generally show increases. This pattern is reflective of the crucial role of CRM1 in exporting TiLV RNA from the nucleus. Inhibiting CRM1-mediated export has multiple effects on the virus. It leads to alterations in the expression of viral proteins, disrupts viral replication, and hinders the complete assembly of viral particles. As a result, the infectivity of the virus is reduced. Additionally, inhibiting CRM1-mediated export triggers a stronger antiviral immune response in the host, enhancing its ability to fight against the virus [8]. CRM1 binds to nuclear export sequences (NES)-cargoes in the nucleus via a cluster of leucine-rich or hydrophobic amino acids in the presence of RanGTP. The tri-complex (RanGTP) is hydrolyzed by RanGAP (a GTPase) resulting to RanGDP after translocation into the cytoplasm. After being released, CRM1 protein returns to the nucleus, and this cycle is repeated to take advantage of or regulate CRM1-mediated nuclear export [9]. A CRM1-targeting therapeutics drug has the potential to be precisely effective against TiLV, thereby inhibiting damage from the infection. Additionally, the compound PKF050-638 was first discovered as a suppressor of the HIV Rev protein; however, it also disrupts the activity of CRM1. The N-azolylacrylate scaffold was subsequently used by the SINE compounds, a group of small-molecule inhibitors that have a similar structure. This group includes KPT-185, KPT-276, KPT-335, KPT-330 (selinexor), KPT-8602 (eltanexor), and SL-801 (felezonexor). The SINE compounds have a unique affinity for Cys528 inside the cargo-binding groove of CRM1 [6].

It is imperative to discover viable phytochemicals and small bioactive molecules for drug development against infectious viral diseases in aquaculture through the investigation of novel compound interactions with the targeted viral protein [10]. Computational or *in silico* methods are assisting in making decisions and simulating virtually every aspect of drug discovery and development, which is a rapid and straightforward method for predicting liable compounds against certain drug targets [11]. Moreover, those compounds can be selected, documented and scored for their interactions based on molecular docking. The ADMET properties, which include absorption, distribution, metabolism, excretion, and toxicity, play a decisive role in determining the efficacy and safety of compounds in pharmacokinetics. Computer-aided methods can be employed for straightforward prediction of these properties. Additionally, MD simulation can be utilized to confirm potential drug candidate stability when interacting with the targeted protein [12].

Several marine and terrestrial plants have been discovered to contain bioactive compounds possessing antiviral drug properties [13]. Natural phytocompounds are increasingly being used as antimicrobial drugs in aquaculture instead of synthetic antibiotics [14]. Although it is challenging to develop novel drugs, advancing investigations for the discovery of bioactive compounds that target specific protein of infectious aquatic disease etiologies is urgently needed [15]. Previously, mangrove plants and medicinal plant garlic (*Allium sativum*) derived natural bioactive ligand was used against Nervous Necrosis Virus (NNV) capsid protein and *in-silico* study revealed the capacity of the ligand to inhibit this viral replication in Asian Seabass, *Lates calcarifer* [16]. Similarly, amphene is the best-screened phytochemical with the lowest binding energy with MX protein complex that can inhibit NNV infectivity in this species. Furthermore, a computational drug development approach revealed that phytocompounds, namely friedlein, phytosterols and 1-Triacontanol from *Avicennia alba* could bind to viral gag polyprotein and inhibit infectious Walleye Dermal Sarcoma Virus (WDSV) replication [15].

In recent decades, mangrove plants have been documented as a source of new chemical products with a rich assortment of natural bioactive properties. Mangrove plants, *Heritiera fomes* and *Ceriops candolleana* have a long history of use for medicinal and other purposes (making pickles, nuts, boats etc.) by people in coastal populations. Several studies have revealed that root, bark and leaves of both plants possess significant antioxidant, antinociceptive, antihyperglycemic, antimicrobial (such as antibacterial, antiviral and antifungal), and anticancer properties [17]. Compounds such as 10-undecenoic acid, methyl ester hexadecane dibutyl phthalate hexadecanoic acid, methyl ester undecylenic acid 9,12-octadecadienoic acid (Z,Z)-, methyl ester palmitic acid 2,5-anhydrogluconic acid aminopyrine procyanidins, phytosterols, 9-epiquinine, AC1L1UKB bis (acetic acid), tannins, cholesterol, delta 7-avenasterol, and squalene have been isolated from different parts of these two plants including root, bark and leaves, with impressive potential for a wide range of pharmaceutical products as replication inhibitors for various types of viruses, including TiLV [18].

To evaluate the potential of a new drug candidate against the TiLV virus, this research employed various readily available *in-silico* methodologies. These approaches included virtual screening, molecular docking, homology modeling, QM calculation, ADMET analysis, and MD simulation. The objective was to investigate the effectiveness of the drug candidate in targeting either CRM1 or XPO1 proteins associated with the TiLV virus.

## Materials and methods

### Phytochemical and protein preparation

The phytochemicals from the *C. candolleana* were sourced from the IMPPAT database, while those from *H. fomes* were obtained from published literature [19]. In total, 17 compounds

from *C. candolleana* and *H. fomes* were collected and stored in a 2D (SDF) file format. To prepare ligands for further analysis, the phytochemicals were optimized and converted into pdbqt files. Similarly for the docking process, optimization and preparation of the proteins via Auto-Dock tool was conducted, and its corresponding file was saved in the pdbqt format.

## Protein structure prediction through homology modeling

To obtain the TiLV structural CRM1 fasta sequence (NC_031978.2), the NCBI database (https://www.ncbi.nlm.nih.gov/gene/100703095, 30 April 2022) was exploited and submitted to the popular online web portal (https://zhanglab.dcmb.med.umich.edu/I-TASSER/, on 30 April 2022), viz. Iterative Threading Assembly Refinement (I-TASSER) for confirmation or prediction of the 3D structure of the anticipated protein. The C-score, TM-score value, and root mean square deviation (RMSD) of the top five models of that structure were then provided by the I-TASSER [20].

## Protein structure refinement and validation

The 3D protein structure was refined by uploading it online to GalaxyRefine (http://galaxy. seoklab.org/cgi-bin/submit.cgi?type=REFINE, 7 May 2022), from which the data about the value of RMSD, energy score, and final structure quality were obtained. To select the refined structure, several parameters including RMSD, average distance among atoms and energy score were used. The refined structure was pictured using PyMol v2.3.4 software. For validation of the refined model, the Ramachandran plot scoring function was used [21].

## Preparation of protein and ligand

Protein structures need to be refined and further validated before docking can take place. Thus, the selected 3D structure of CRM1 was generated using AutoDockTools (ADT). Nonpolar hydrogen atoms were combined, polar hydrogen atoms were introduced, and the Gasteiger charges of the protein were computed [22]. Additionally, it served as a cofactor and metal ion discarded from its protein. An extensive literature review and the database (https://cb.imsc.res. in/imppat/home) of Indian Medicinal Plant, Phytochemistry, and Therapeutics (IMPPAT) were used to identify the selected botanical compounds. Seventeen compounds were retrieved, among which 7 compounds were identified from *C. candolleana* and 10 compounds from *H. fomes*. To set and minimize energy for compounds selected from the database, The Universal Force Field (UFF) was used, as specifically designed for each ligand.

## Identification of binding site and grid box generation

The refined protein structure of CRM1 was submitted to the CASTp 3.0 server (http://sts.bioe. uic.edu/, 10 May 2022) to predict the active site residues. Then, the server identified different active pockets, and the first of these was selected based on the pocket surface area (SA) and volume. The active pocket and corresponding AA residues were retrieved to visualize the binding pocket of the protein through the BIOVA Discovery Studio Visualizer Tool 16.1.0. The residues of the binding sites were then generated by BIOVA, used to select the grid box for molecular docking simulations [23].

## Molecular docking simulation

PyRx software was used to run virtual screenings of selected compounds with analyses of the molecular docking process. The virtual screening software identified several hypothetical drugs to treat several diseases. In addition to the Lamarckian genetic algorithm (LGA) for scoring,

AutoDock and AutoDock Vina for docking were provided. This study employed PyRx tools AutoDock Vina to perform docking interactions between molecules. For the analysis of complex binding poses, the BIOVA Discovery Studio Visualizer Tools 16.1.0.41 was used [24].

## ADME analysis

During the preliminary drug development phase, an ADME assessment is required to determine a drug's safety and efficacy in living systems. ADME properties depict pharmacokinetic behavior to predict the movement of drugs through and clearance out of the body. This study utilized the SwissADME (http://www.swissadme.ch, 11 May 2022) to interpret the response of selected drug candidates regarding pharmacodynamics [25].

## Toxicity test

It is economically advantageous and vitally important to conduct toxicology tests during the drug development process in order to determine whether the compound has toxic properties, as well as for the assessment of dosage requirements. To assess the toxicity of chosen chemicals, *in-silico* computational approaches were utilized to evaluate their safety profiles. For early-stage toxicity evaluation, ProTox-II was used (http://tox.charite.de/protox_II, 11 May 2022), which simulated characteristics associated with the compounds' acute toxicity, carcinogenicity, hepatotoxicity, mutagenicity, cytotoxicity, and immunotoxicity [26]. The investigators chose to use the Toxicity Estimation Software Tool (TEST) for the purpose of evaluating the toxicity of the chemical under investigation, eliminating the need for supplementary software. The use of quantitative structure-activity relationship (QSAR) methodologies has been employed to provide estimations for the chosen compounds. The pathways addressed by this website span a broad spectrum, including those associated with nuclear receptor signaling as well as those involved in the physiological response to stress [27].

## Molecular dynamic (MD) simulation

The thermodynamic stability of the receptor-ligand system was evaluated using the Desmond v3.6 Program, which facilitated the automated simulation and free energy perturbation (FEP) calculations at different temperatures to predict the equation of state (EOS). Orthorhombic periodic boundary conditions were applied to simulate larger complex structures. To stabilize the system and achieve energetic neutralization, sodium ions were introduced and randomly placed [28]. MD simulations were conducted using the buffer box calculation method with a size of 10 Å and a specific box shape. Periodic boundary conditions and the NPT ensemble were employed during the simulations. Trajectories at 0 ns, 50 ns, and 100 ns were extracted for each ligand-CRM1 complex and then aligned. The retrieval and analysis of the resulting trajectory from the simulations was done using the Simulation Interaction Diagram (SID) in Schrödinger Release 2020–2.

## MM-GBSA analysis

The free energy associated with the binding of chemicals and proteins was computed. The system under consideration is characterized by its complexity, and the score is determined by evaluating the free energy of binding based on the trajectory obtained from a molecular dynamics (MD) simulation. The MM-GBSA method was used to determine the binding free energy (Gbind) between certain compounds and the importin-11 protein. The Schrodinger Maestro software program was used for the computation of MM-GBSA.

# Results

## Protein 3D structure, refinement and validation

Among the models of CRM1 provided by I-TASSER, the best 3D protein structure was selected based on the lowest C-score (-0.01). Protein model-2 was picked following refinement, which had a 3D refined score of 37759.1, a GDTHA score of 0.9666, an RMSD value of 0.355, and no MolProbity issues. Prior to refinement, the Ramachandran plot analysis of CRM1 revealed that 97.629% of residues were in the favorable region, 2.062% were in the allowed region, and 0.309% were in the disallowed region. After refinement, the refined CRM1 model showed improved results, with 97.984% of residues in the favorable region, 2.016% in the allowed region, and 0.000% in the disallowed region (S1 Fig).

## Identification of active site and receptor grid generation

The study first identified CRM1's active side (AS) from CASTpi, then obtained its combined binding location (Fig 1). The protein binding site residue was identified through active pocket analysis (Fig 1). Four active pockets were selected from a total of 197 active pockets based on their surface area. The chosen active pockets and their corresponding amino acid residues are represented as balls in Fig 1. During the simulation procedure of molecular docking, the server-identified binding sites were used to build a receptor grid with grid box dimensions of X = 77.1430, Y = 81.9808, and Z = 93.4587 in angstroms (Å).

## Molecular docking

After molecular docking of phytochemicals compounds, the binding affinity was determined to have varied between −3.2 and −8.7. The top 20% of the 17 phytochemicals with the highest binding affinity were selected. The top three compounds, specifically procyanidins (CID107876), delta7-Avenasterol (CID 12795736), and phytosterols (CID12303662), were chosen according to their docking scores respectively of −8.3, −8.2 and −7.9 kcal/mol. In addition, the commercially marketed drug Leptomycin B (docking score -6.6) was chosen and all four compounds were examined using various screening techniques. The three most favorably-ranked compounds scores are enumerated in Table 1, while docking scores are recorded in S1 Table for all the phytochemicals. Validation of those scores for the three selected compounds were confirmed by re-docking. The single-binding poses of three preselected phytochemicals were obtained and re-docked at the same binding site as before (S3 Table and S3 Fig). The RMSD values for the upper and lower ranges were determined, and it was found that the binding affinities of the phytochemicals remained similar to the earlier scores (S3 Table). The following four phytochemicals, CID107876, CID12795736, CID12303662 and CID6917907 with docking scores respectively of -8.3, -8.2–7.9 and -6.6 kcal/mol and zero (0) RMSD value were chosen. The root mean square deviations (RMSDs) of the expected postures from the observed binding modes for FlexX were calculated. The vast majority of the near solutions in this software have root mean square (RMS) deviations that are below 1.5 Å. The majority of active site postures may be classified within the range of 1.0 to 3.5 Root Mean Square (RMS) values. Nevertheless, it is evident that many active site solutions within the range of 0 to 2.0 rms were deemed to be near poses in the subjective analysis. This observation may be made by comparing the cumulative values for close assignments in S4 Table.

List of Compounds, PubChem CID, Chemical Name, Molecular formula, molecular weight, and docking score of the seventeen selected compounds with the highest binding affinity are shown in S2 Table.

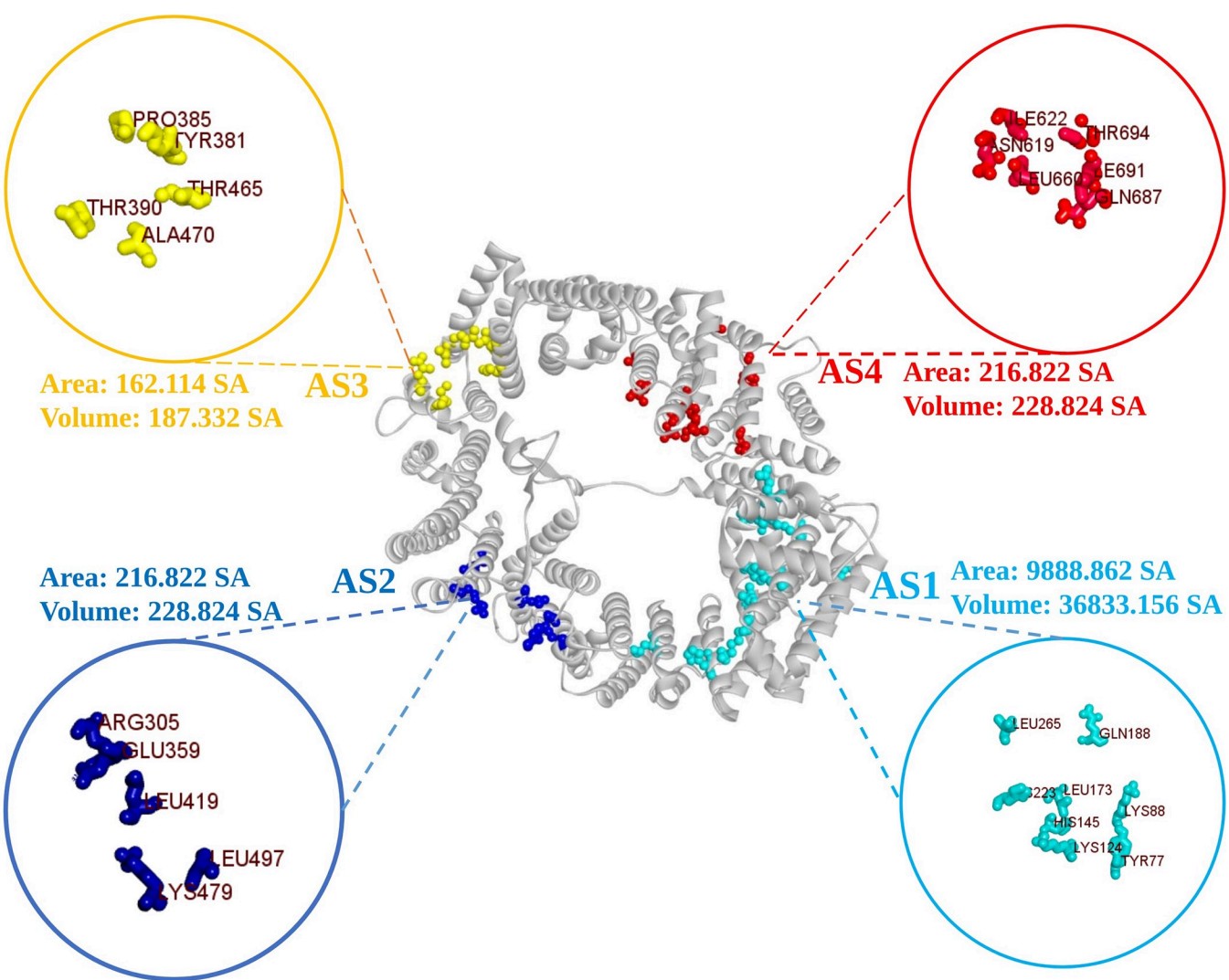

**Fig 1. The surface area of the active pockets of CRM1 (selected four) was calculated using the CASTp server.** All the active sites i.e., AS1, AS2, AS3 & AS4 with their corresponding amino acids are depicted respectively in light blue, deep blue, yellow and red.

## Protein-ligand interaction analysis

Compounds that have the highest binding scores were chosen to see how they could interact with CRM1. Several hydrogen and hydrophobic bonds were reported to form between the compound CID107876 and the desired CRM1 (Fig 2). The hydrogen bonds were discovered to form at PRO427 and LYS590 position, while the hydrophobic bonds form at the location

**Table 1. The best three phytochemicals with their name, formula and CID.**

| PubChem ID | Compounds | Molecular Formula | Molecular Weight | Docking Score (kcal/mol) |
|---|---|---|---|---|
| 107876 | Procyanidin | C30H26O13 | 594.5 | -8.3 |
| 12795736 | delta7-Avenasterol | C29H48O | 412.7 | -8.2 |
| 12303662 | Phytosterols | C29H50O | 414.7 | -7.9 |
| 6917907 | Leptomycin B | C33H48O6 | 540.7 | -6.6 |

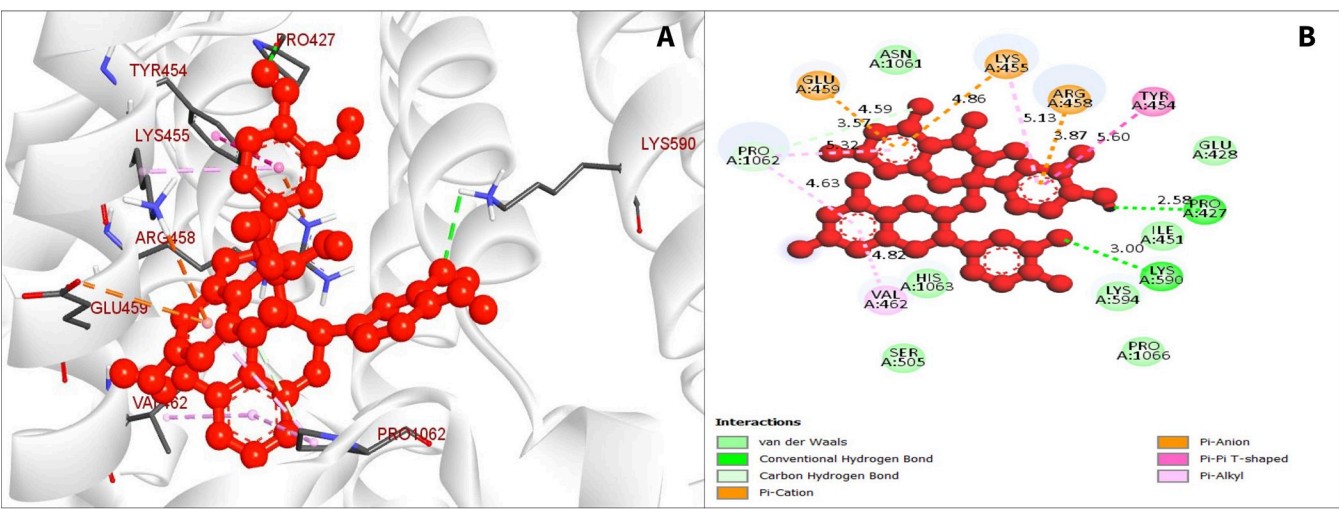

**Fig 2. The interaction between the chemical CID107876 and CRM1.** (A). The 3D interaction is shown on the left of the protein ligands, while the 2D interaction is shown on the right (B).

TYR454, VAL465, PRO1062, PRO1062, PRO1062, GLU459, ARG458, LYS455, and LYS455 position, as illustrated in Fig 3 and the bond types in Table 2.

The compound CID12795736 has formed several bonds in the VAL120, ILE127, LYS124, and VAL169 residual position, which were hydrophobic. Another two conventional hydrogen bonds were formed at the position of SER162 AA position as listed in Fig 3 and illustrated in Fig 4.

For compound CID12303662, also various hydrophobic bonds occurred with desired phytochemicals, with alkyl bonds in VAL169, LYS124, ILE127, and VAL120 residual positions as depicted in Fig 4 (and as listed in Table 2).

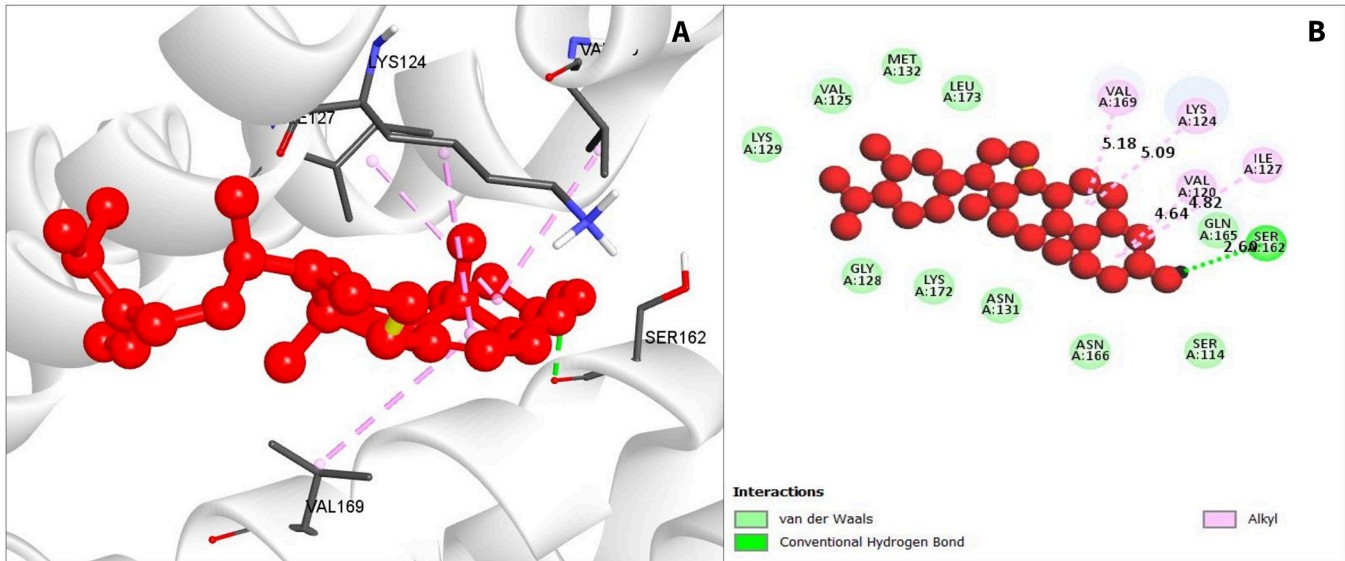

**Fig 3. The interaction among chemical CID12795736 and CRM1.** (A). The 3D interaction is shown on the left of the protein ligands, while 2D interaction is shown on the right (B).

**Table 2. Compilation of bonding interactions among three chosen phytochemicals and CRM1.**

| PubChem ID | Residue | Distance | Category | Type |
|---|---|---|---|---|
| CID 107876 | PRO427 | 2.58 | Hydrogen Bond (HB) | Conv-H-Bond |
| | LYS590 | 3 | HB | Conv-H-Bond |
| | TYR454 | 5.6 | Hydrophobic (HP) | Pi-Pi- T-Shaped |
| | VAL465 | 4.82 | HP | Pi-Alkyl |
| | PRO1062 | 4.63 | HP | Carbon-H-Bond |
| | PRO1062 | 3.57 | HP | Carbon-H-Bond |
| | PRO1062 | 5.32 | HP | Carbon-H-Bond |
| | GLU459 | 4.59 | HP | Pi-Cation |
| | ARG458 | 3.87 | HP | Pi-Cation |
| | LYS455 | 4.86 | HP | Pi-Cation |
| | LYS455 | 5.13 | HP | Pi-Cation |
| CID 12795736 | SER162 | 2.6 | HB | Conv-H-Bond |
| | VAL120 | 4.64 | HP | Alkyl |
| | ILE127 | 4.82 | HP | Alkyl |
| | LYS124 | 5.09 | HP | Alkyl |
| | VAL169 | 5.18 | HP | Alkyl |
| CID 12303662 | VAL169 | 4.6 | HP | Alkyl |
| | LYS124 | 5.07 | HP | Alkyl |
| | ILE127 | 4.7 | HP | Alkyl |
| | VAL120 | 4.78 | HP | Alkyl |

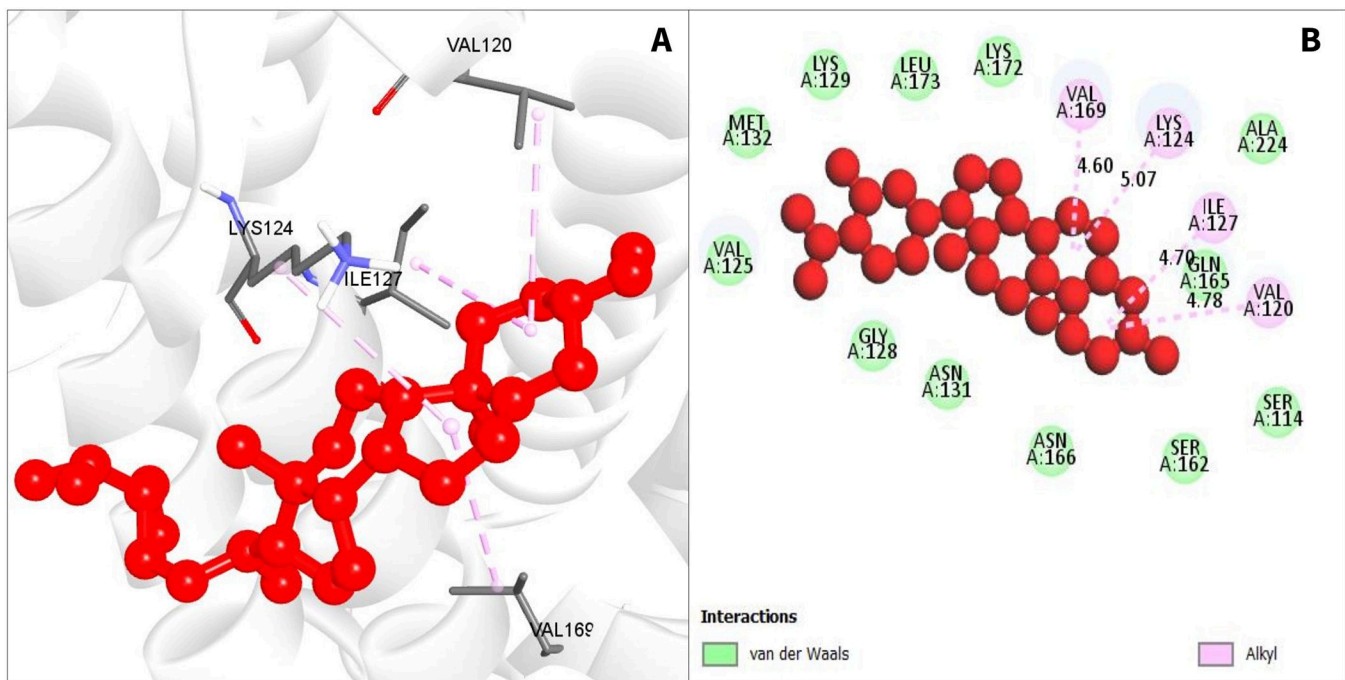

**Fig 4. Interaction among chemical CID12303662 and CRM1.** (A). The 3D interaction is shown on the left of the protein ligands, while 2D interaction is shown on the right (B).

**Table 3. The pharmacokinetics list comprises the ADME features of the three chosen drugs and various physicochemical features of these substances.**

| Properties | Parameters | CID 107876 | CID 12795736 | CID 12303662 |
|---|---|---|---|---|
| | MW (g/mol) | 594.52 g/mol | 412.69 g/mol | 414.71 g/mol |
| | Heavy atoms | 43 | 30 | 30 |
| | Arom. Heavy atoms | 24 | 0 | 0 |
| | Rotatable bonds | 4 | 5 | 6 |
| | H-bond acceptors | 13 | 1 | 1 |
| | H-bond donor | 10 | 1 | 1 |
| | Molar Refractivity | 147.52 | 135.75 | 133.23 |
| Lipophilicity | Log Po/w | 1.98 | 7.94 | 8.02 |
| Water solubility | Log S (ESOL) | -4.9 | -7.48 | -7.9 |
| Pharmacokinetics | GI absorption | Low | Low | Low |
| Drug likeness | Lipinski, Violation | No | Yes | Yes |
| Medi. Chemistry | Synth. accessibility | 5.64 | 6.03 | 6.3 |

Medi. Chemistry = Medicinal chemistry; Synth. Accessibility = Synthetic accessibility.

## ADME analysis

The server SwissADME was used to evaluate the pharmacophore characteristics of the three chosen druglike compounds. The druglike compounds possess lipophilic characteristics which enable them to be dissolved in fats, oils, and other nonpolar solvents. Pharmacophore features demonstrated that the compound is effective and druggable, thus it may be employed in the research. The pharmacokinetic parameters discovered for the three chosen drugs are presented in Table 3. The ADME properties of the chosen three phytochemicals were found to be of high quality.

## Toxicity prediction

The three compounds PubChem CID107876, CID12795736, and CID12303662 were selected previously through various screening processes. The compounds were subjected to analysis in ProTox-II server, which assessed their cytotoxicity, hepatotoxicity, oral toxicity, mutagenicity and carcinogenicity. However, Table 4 shows the phytochemicals that lack oral or organ toxicity effects. After toxicity testing, the results confirmed that the three selected compounds were either non-toxic or exhibited low toxicity levels. Moreover, Comparative Molecular Field Analysis (CoMFA) is a widely used three-dimensional quantitative structure-activity relationship (3D-QSAR) technique. The process of constructing a three-dimensional grid encompassing a molecule of interest and then evaluating the steric (shape) and electrostatic characteristics at each grid point is necessary.

CoMFA result can provide valuable insights into the structure-activity relationship of the studied molecules. This information can aid in the design of novel compounds with the desired activity profiles by assisting researchers in gaining a better understanding of the factors that influence the biological activity of the molecules. The spatial arrangement of the four

**Table 4. The drug-induced toxicity profile of selected phytochemicals.**

| PubChem ID | Hepatotoxicity | Carcinogenicity | Immunotoxicity | Mutagenicity | Cytotoxicity |
|---|---|---|---|---|---|
| CID 107876 | Inactive | No | Inactive | Inactive | Inactive |
| CID 12795736 | Inactive | No | Light active | Inactive | Inactive |
| CID12303662 | Inactive | No | Active | Inactive | Inactive |

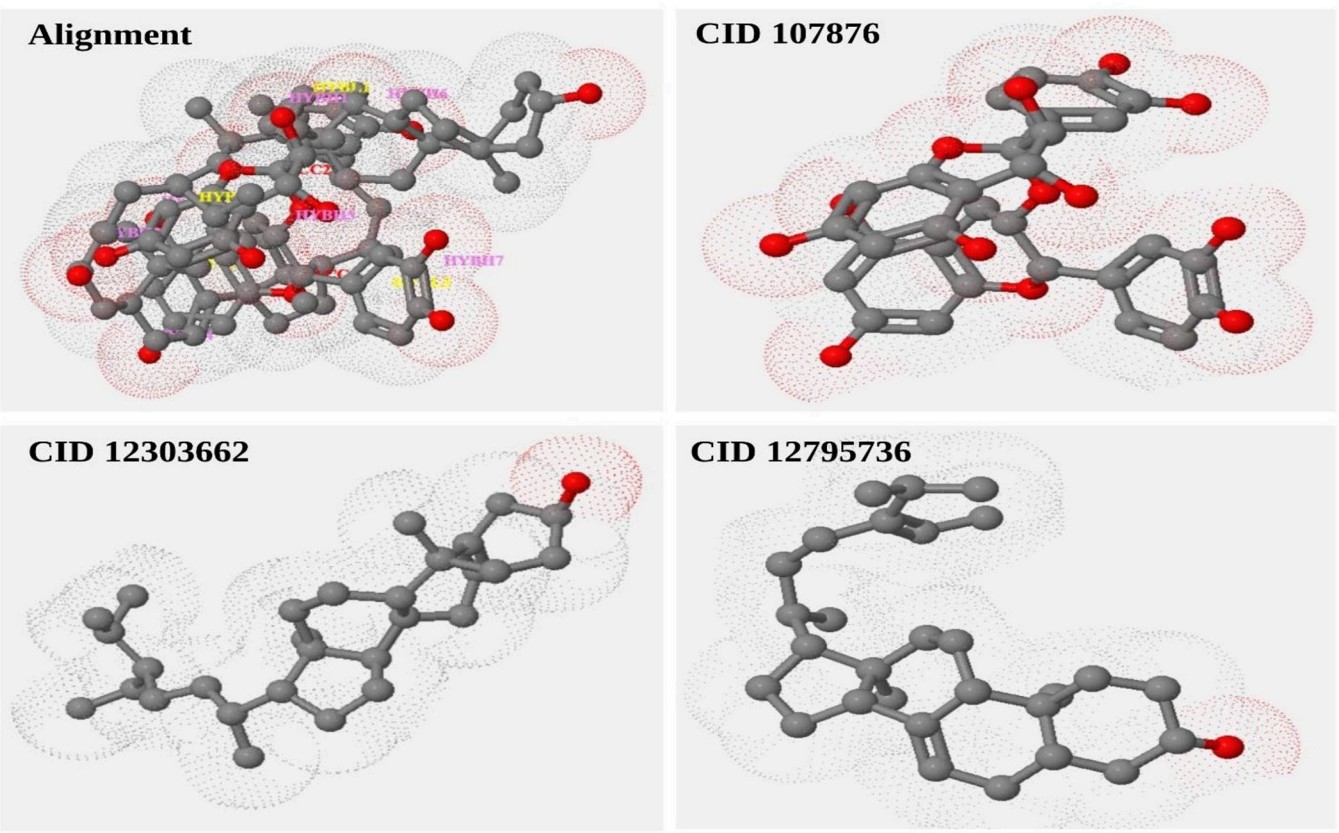

**Fig 5. Construction of a three-dimensional grid encompassing a molecule of interest.** Subsequently, the steric (shape) and electrostatic characteristics are computed at each grid point using the Comparative Molecular Field Analysis (CoMFA) technique. This analysis is performed for a set consisting of the three chosen compounds.

finest compounds results in steric effects. Three compounds, including CID107876, CID12303662, CID12795736 and their alignment were depicted in Fig 5. (S5 Table).

## MD simulation analysis

To investigate the constancy of the protein-ligand docking complexes, MD simulations were carried out. This simulation method offers valuable insights into the intermolecular interactions over time. A 100 ns simulation of MD was performed for intermolecular interactions among protein and ligand complex structure confirmation. The simulated trajectories were then utilized to assess parameters such as RMSD, Root Mean Square Fluctuation (RMSF), mapping of protein-ligand interactions, and analysis of ligand torsion. These analyses provided more depth in our understanding of the dynamic behavior and stability of the protein-ligand complexes.

## RMSD

The RMSD values of three chemicals and the protein were analyzed to assess system equilibration. Specifically, the RMSD values of compounds CID107876, CID12795736, and CID12303662 were compared with the structure of CRM1, as shown in Fig 6, to identify any variations. The RMSD values for all compounds fell within the range of 1 Å to 2.5 Å, which was considered acceptable when contrasted to the CRM1 structure. Notably, the compound

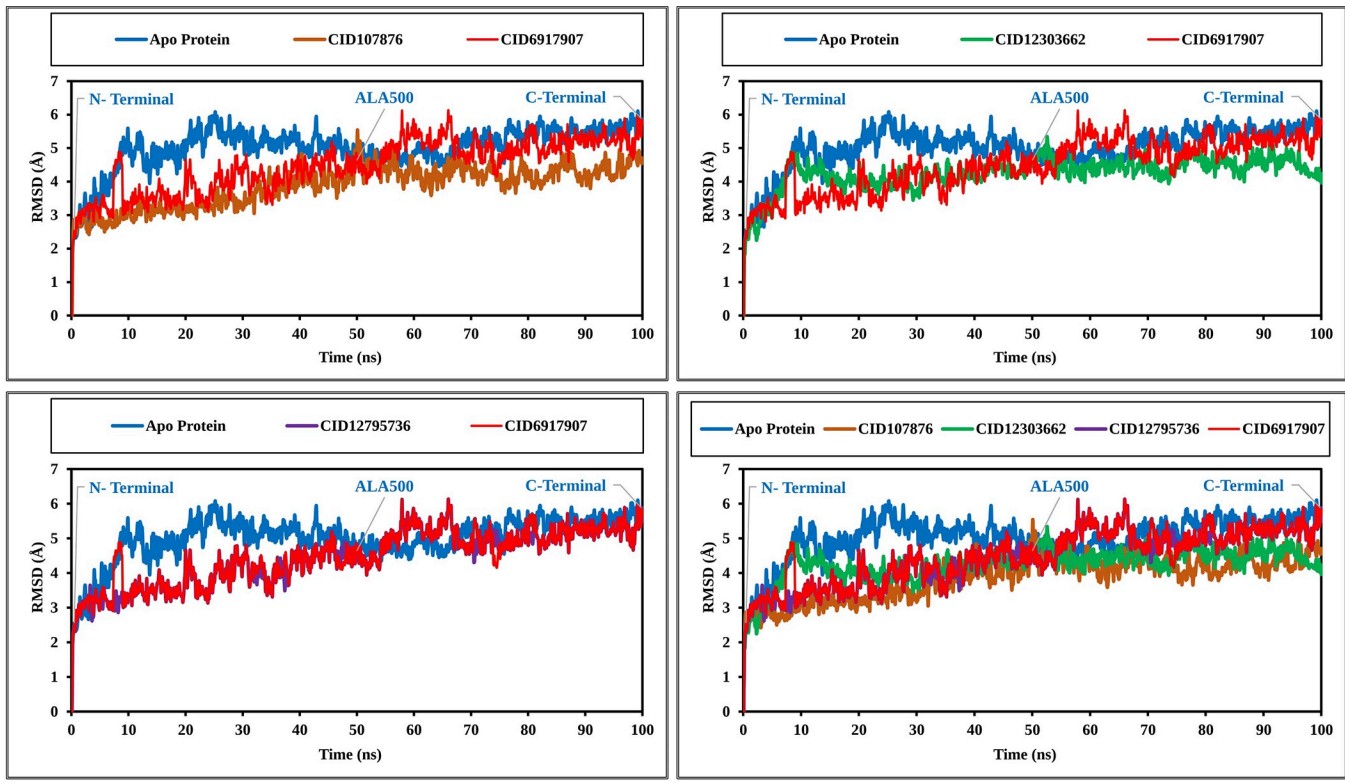

**Fig 6. RMSD values obtained from the Cα atoms of CRM1 (represented by blue curves).** Naturally occurring compounds were analyzed over a 100 ns simulation time. Specifically, the RMSD values for CID107876 are depicted as brown curves, CID12795736 as violet curves, CID6917907 as red curves and CID12303662 as green curves.

CID12795736 exhibited the highest stability between 65 ns to 100 ns simulation time. Additionally, compounds CID107876 and CID12303662 displayed initial fluctuations but gradually stabilized over time. The simulation was observed to converge between 40 ns and 100 ns for all compounds, with the RMSD values reaching a stable value (Fig 6). The marketable drug Leptomycin B was compared with other compounds, which fluctuated from 55 to 68 ns within the period. Consequently, the selected phytochemicals are deemed stable for the targeted protein. The root mean square deviations (RMSDs) of the sole ligand, expressed in units of nanoseconds (ns), are being considered in this analysis. The ligand-hiMGAM snapshots of three phytochemical substances (Phytosterols, Avenasterol, and Procyanidin) and a marketable medication (Leptomycin B) were overlaid at the start and end timeframes. The ligands and bound hiMGAM proteins are visually distinguished by the colours green and red, respectively, in relation to the extracted frames at 0 ns and 100 ns (S3 Fig).

**RMSF.** To assess the displacement of specific atoms during the simulation, analyses of three RMSF complex structures were carried out. These values were further compared to the RMSF of the CRM1 structure, as shown in Fig 7, to observe atomic alterations. The highest peak of fluctuation was observed between residue points 184 and 396. Additionally, the entire complex structure exhibited fluctuation between residues 500 and 550, which was attributed to the location of N-terminal domain. Generally, the fluctuation of all the phytochemicals was found to be more optimal than the RMSF of CRM1. This indicates that the phytochemicals are capable of maintaining a steady interaction with no significant alteration of protein structure.

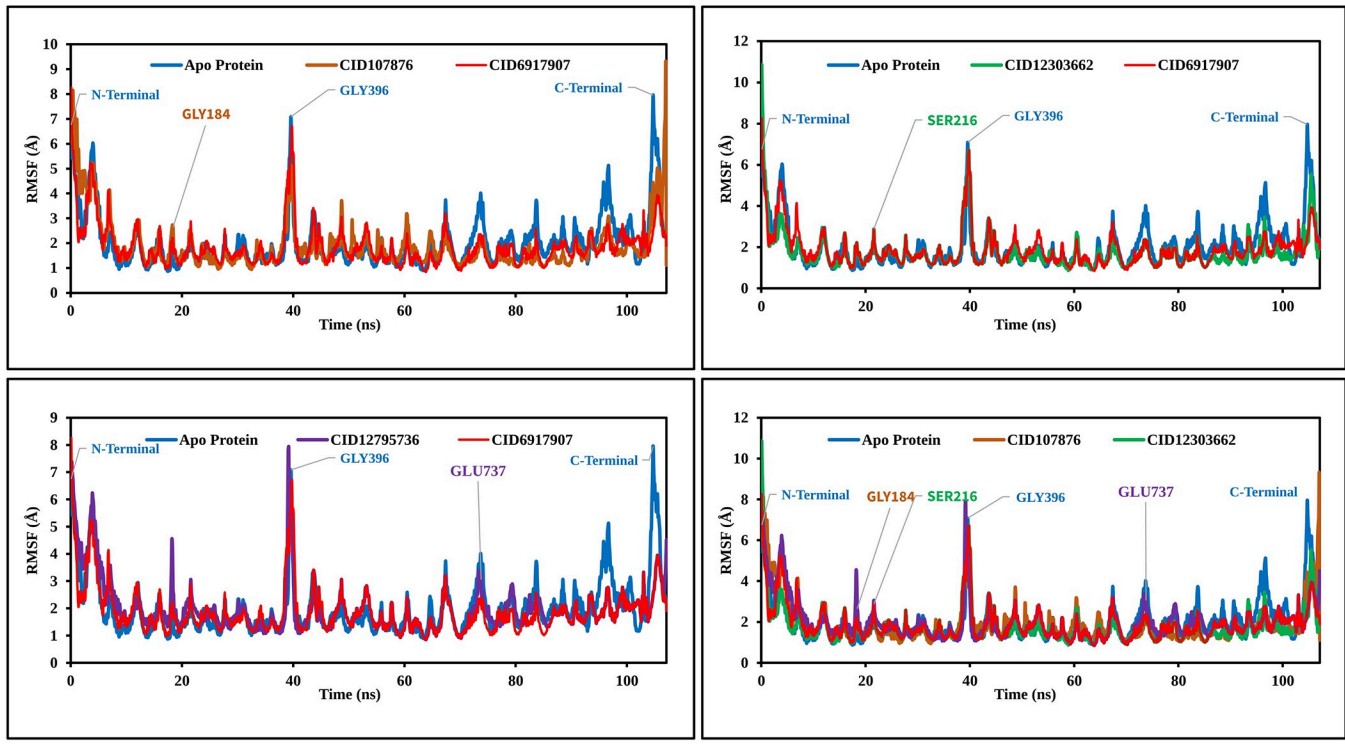

**Fig 7. RMSF values extracted from the protein residue index Cα atoms of the complex structure.** Values shown include CID107876 (brown), CID12795736 (violet), CID12303662 (green), CID6917907 (red), and CRM1 (blue) regarding 100 ns of simulation time.

## Protein-ligand contact mapping

Throughout the analysis of the 100ns MD simulation trajectory. The interaction between CRM1 and the three selected phytochemicals (CID107876, CID12795736, and CID12303662) were monitored. These interactions were categorized as hydrogen bonds, hydrophobic bonds, ionic bonds and water bridge bonds. The stacked bar graphs in Fig 8 show the summary of those interactions. All three of the selected compounds had a strong interaction with catalytic residues. The molecular dynamic simulation intervals of the screened compounds were found to contain a significant amount of intermolecular interaction. Active residues were discovered to interact with intermolecular and chosen ligands, and their density was determined for each of the three compounds studied. The active pocket of CRM1 was identified by the screening of natural chemicals and the study of MD trajectories. In real-life experiments, therefore, the screened natural compounds CID107876, CID12795736 and CID12303662 can be predicted to have a high degree of stability with the CRM1.

## Ligands torsion analysis

The ligand torsions plot provides a summary of the conformational changes observed in each rotatable bond (RB) of the ligand during the entire simulation trajectory (0-100ns), as depicted in Fig 9. In the upper section, a two-dimensional representation of the ligand is shown, highlighting the RB (torsion) regions using different colors. Each RB torsion is accompanied by a dial plot and corresponding bar plots in the same color. The dial plots depict the changes in conformation of the torsions during the simulation. The center of the radial plot represents the beginning of the simulation, while the radial direction indicates the passage of time. The

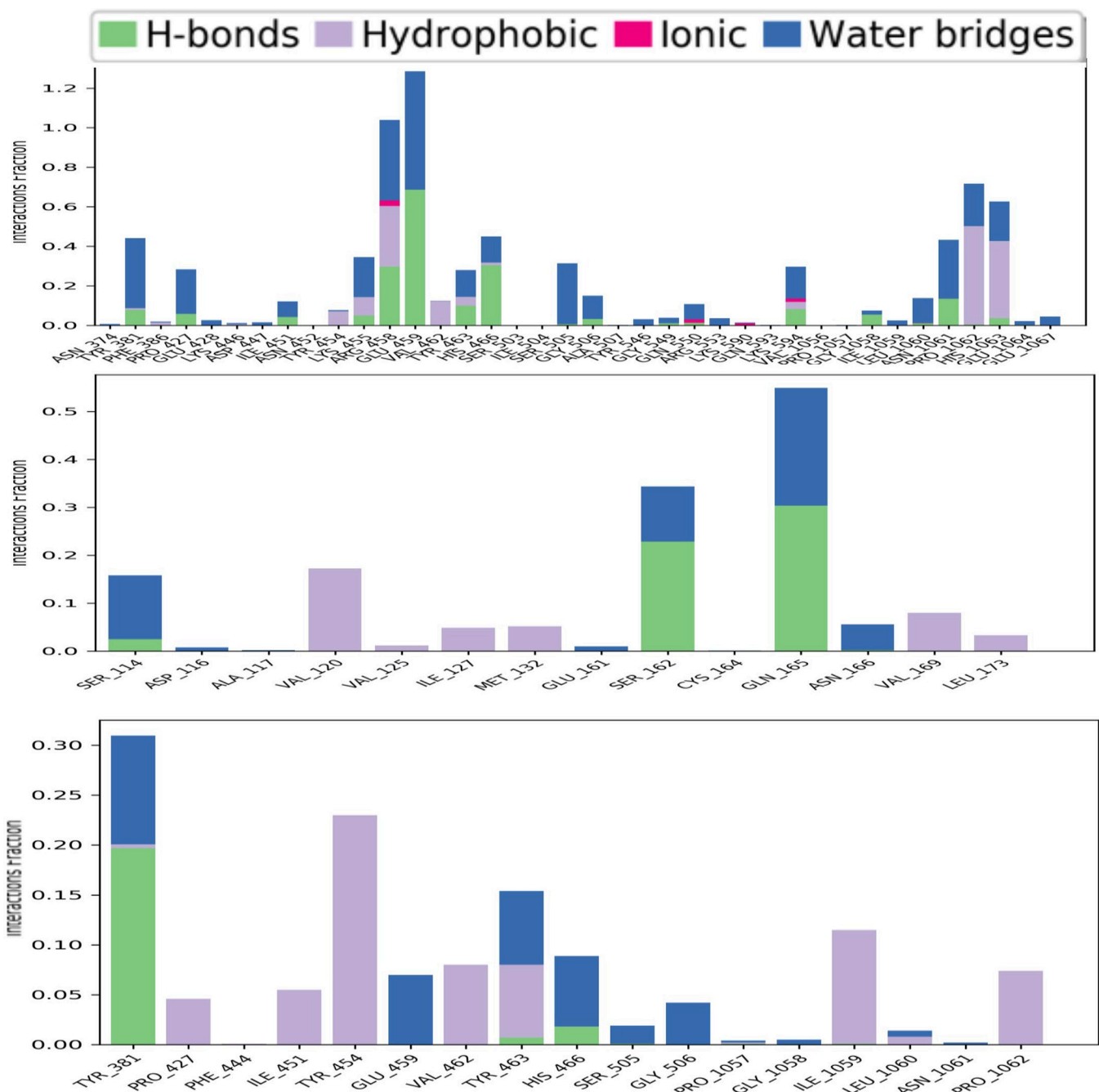

**Fig 8. Stacked bars of the contact mapping of CRM1 with potential natural phytochemicals.** Illustrated here are (A) CID107876; (B) CID 12795736; and (C) CID 12303662 determined from simulations trajectory of 100 ns.

bar plots provide a summary of the dial plots, showing the probability density of the torsion at different time points. Furthermore, if information about the torsional potential is available, the plot also includes the potential energy of the RB by summing the potential energy of the relevant torsions. The values of the potential energy are displayed on the left Y-axis of the chart and are expressed in kcal/mol. Analyzing the histogram and the relationship between torsional

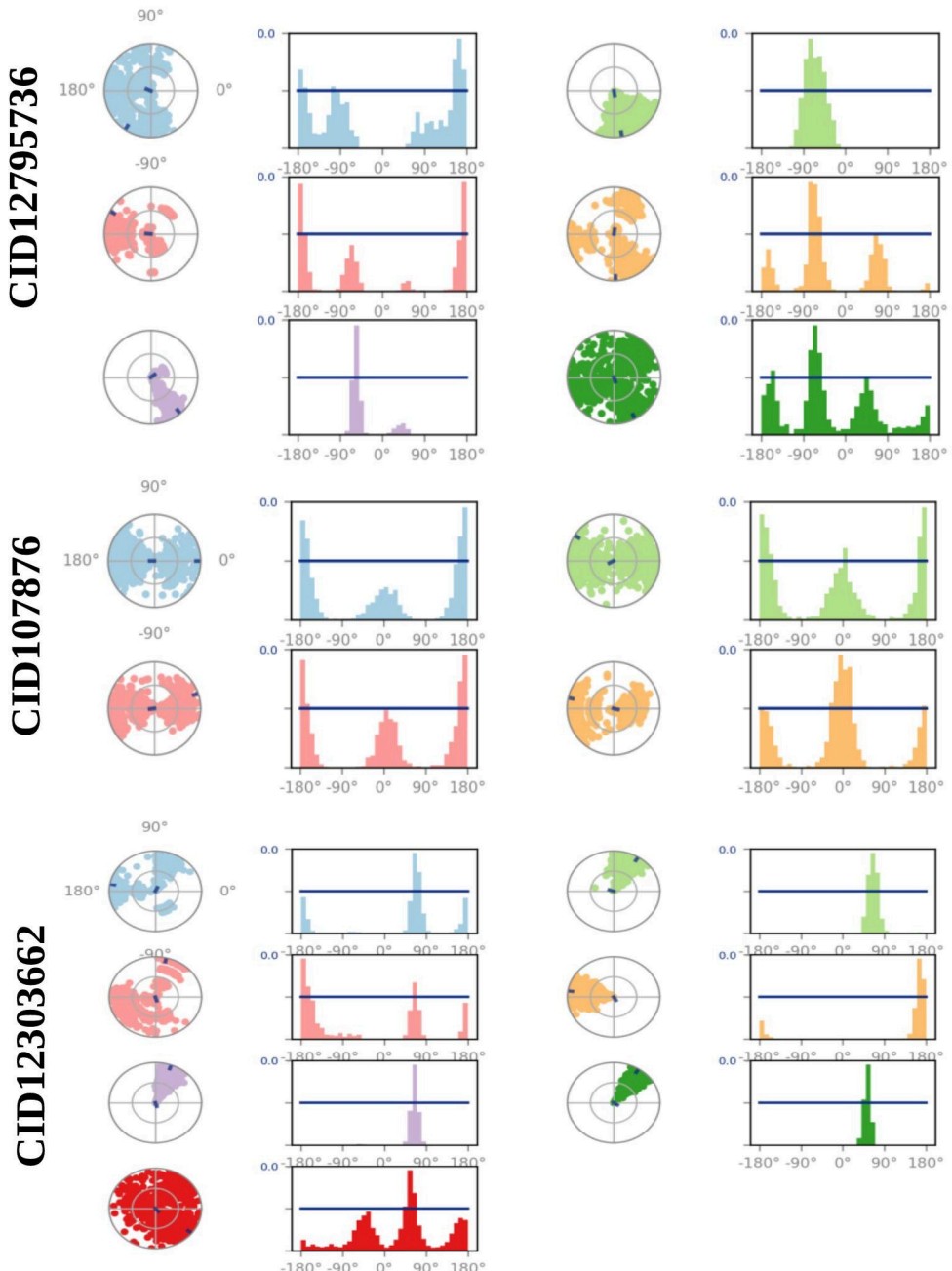

**Fig 9. Ligand torsions plots.** These provide a summary of how each RB in the ligand underwent conformational changes throughout the entire simulation trajectory (from 0 to 100 ns).

potential and torsion values can offer insights into the strain experienced by the ligand in maintaining its conformational bond with the protein.

## MM-GBSA analysis

The estimation of the binding free energy between small molecules from mangrove plants and CRM1 has often been conducted by using molecular energies in conjunction with the commonly used Born and surface area continuum solvation methods. The subject matter was

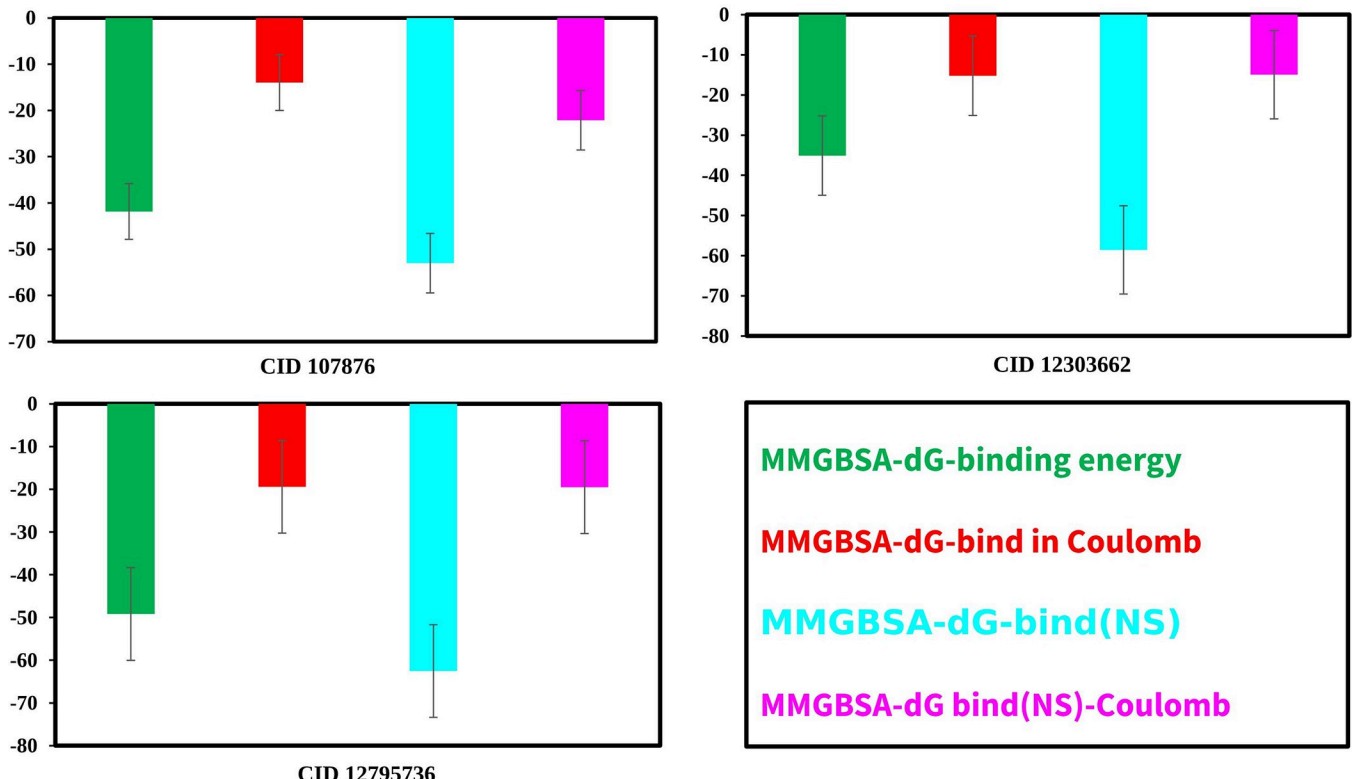

**Fig 10. MM/GBSA calculation from the selected three compounds.** Binding free energy for each of the three compounds in association with CRM1 is given in kcal/mol and in Coulombs. Calculated No Strain binding values (NS) for each pair are also shown in both kcal/mol and Coulombs.

positioned within the spectrum of empirical scoring and rigorous alchemical perturbation methodologies (Fig 10). The binding free energy values obtained for the three compounds, namely CID: 107876, CID: 12303662, and CID: 12795736, in complex with CRM1, were determined to be -41.8499±5.08, -49.1706±12.41 and -35.0858±8.19 kcal/mol, respectively (S6 Table).

## Solvent accessible surface area

The solvent-accessible surface area (SASA) has an impact on both the structure and function of biological macromolecules. The active sites of proteins are comprised of amino acid residues on their surfaces, which have the ability to bind ligands. This interaction allows for a better understanding of the hydrophilic or hydrophobic characteristics shown by molecules and protein-ligand complexes, resembling the behavior of solvents. Hence, Fig 11 exhibits the calculated solvent-accessible surface area (SASA) value of the protein while interacting with the chemical compounds (A) CID: 107876, (B) CID: 12303662, and (C) CID: 12795736. In the case of intricate systems, it was shown that the average solvent accessible surface area (SASA) value ranged from 200 to 300 A (Fig 11). This range indicates a substantial level of interaction between the amino acid residue and the molecule under investigation.

## Discussion

Since the emergence of TiLV in 2014, eradication of the pathogen infectivity is the major concern of the world scientific community and virus researchers. So far, no anti-viral drug candidate has emerged that can effectively counteract these zoonotic pathogens. It has been found

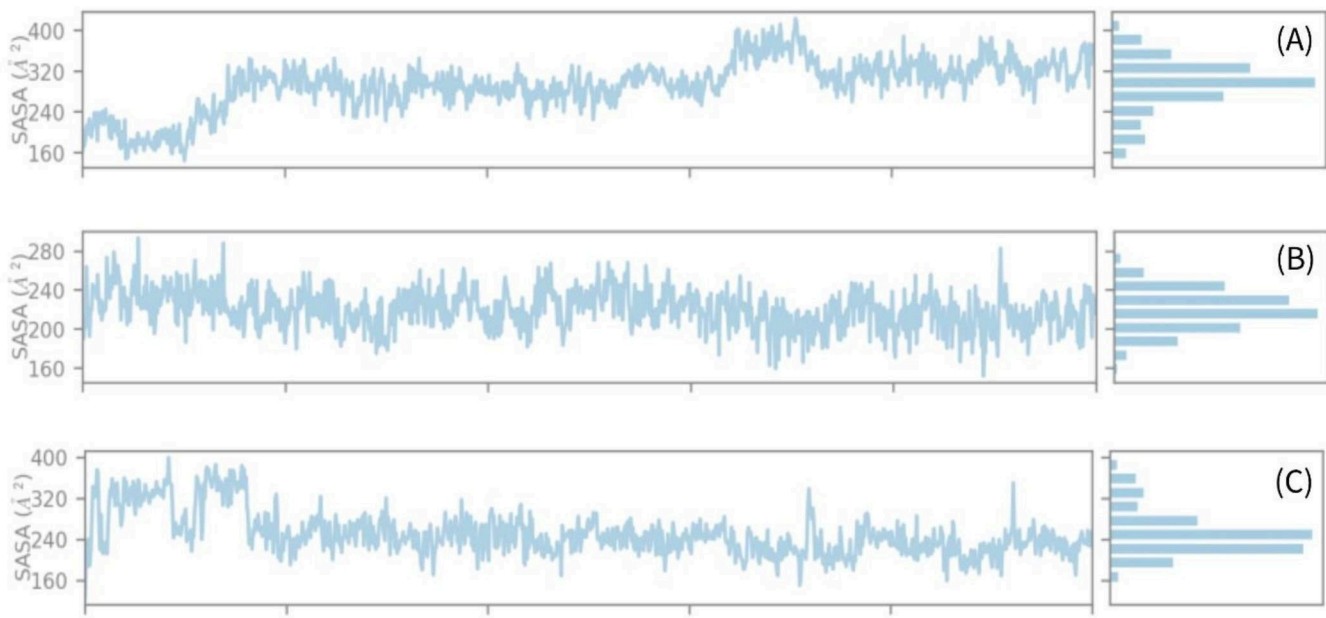

**Fig 11. Solvent accessible surface area (SASA) of the protein–ligand complex.** These values were calculated from the compounds (A) CID: 107876, (B) CID: 12303662, and (C) CID: 12795736 until 100 ns simulation.

that TiLV's protein 4 functions when CRM1 is inhibited, which plays a key role in the export of TiLV RNA [29]. Hence, this study was targeted to inhibit the CRM1 in TiLV through the identification and characterization of a novel and potentially effective bioactive antiviral drug compound. Computer-Aided Drug Design (CADD) is one of the most promising techniques for selecting novel compounds against a specific protein because it comprises a variety of highly advanced and nuanced characteristics and approaches [30]. CADD provides more affordable methods of designing, decreases recruitment costs, and reduces time in a new era of modern drug development. It has gained popularity and acceptance both in academic circles as well as among commercial pharmaceutical professionals. A process involving molecular docking, ADME-Tox prediction, and MD simulations are used to identify drug candidates that are predicted to possess the highest biological efficacy [31]. To understand disease mechanisms and facilitate drug design, it is crucial to identify disease-associated proteins and explore their interactions with potential ligands. By identifying compounds that can interact with and inhibit specific proteins essential for virus replication, it becomes possible to block viral infections and their progression. Through CADD, the specific target molecule can be identified based on its behavior and ligand's binding mode. Additionally, molecular docking identifies the dominant modes of binding within a ligand, as well as protein and MD simulations revealing the mechanisms of protein-ligand interaction [32]. As a result, small-molecule candidates can be recognized as suitable drugs against a particular disease including TiLV infection.

In this study, we first predicted protein structure through Homology Modeling, then refined and validated by utilizing the popular online web portal (including, I-TESSER, GalaxyRefine). A homology-based structural model is widely used in identifying the effects of single nucleotide polymorphisms, as well as in drug design [33]. Homology models contain substantial information about the spatial arrangement of key residues in the protein to investigate the binding site and design/dock drugs that are suited for binding to the molecule. The most highly-ranked protein 3D structure (model 2) with the lowest c-score (-0.01)

generated from I-TESSER was used. To validate the refined model, the Ramachandran plot and z-score value scoring function was utilized. The model quality was increased after refining from the Galaxy Refine server, and the final refined model showed 97.984% in the most favored region in the Ramachandran plot and 0.0% in the disallowed region, indicating good model quality.

Seventeen natural phytochemical compounds were screened, and molecular docking simulations were performed to target the XPO1 to fight and block the replication of TiLV, and among those three most promising compounds were chosen in accord with their highest binding affinity scores for further validation. The compounds with higher docking scores were CID107876, CID12795736 and CID12303662 with binding scores of −8.3, 8.2 and −7.9 kcal/mol, respectively (see Fig 10 and S6 Table). Among these, CID 12795736 is estimated to have the strongest binding affinity with the target protein, but it also displayed the highest standard deviation, suggesting a higher degree of variability or uncertainty in the calculation. The strongest Coulombic interaction with the target protein was apparent for CID 107876, and this compound also displayed the lowest variability in calculated results.

Compounds derived from marine plants have shown antiviral and pharmacological properties against many illnesses. Marine plants and their components are of significant importance in the field of medication design research as well as in the preservation of animal and human life. Many herbal extracts derived from plants have been used to isolate individual molecules that exhibit potential as therapeutic candidates. These molecules are now being employed in many biochemical and biomedical fields [34]. The value obtained for the selected compounds was highly favorable and exhibited characteristic features with favorable indications of their potential as drug candidates. These compounds demonstrated drug-like qualities, as confirmed by their compliance with Lipinski's Rule of Five (RO5). These selected three compounds have been evaluated and all compounds have shown a good value of the ADMET properties. Generally, the ADMET acts on the drug-related properties of pharmacokinetics (PK). Before developing a promising drug candidate, the PK parameters should be optimized since they need to pass standard clinical trials [31]. ADMET properties were traditionally predicted at the end of the drug discovery process, however, *in-silico* tools can predict them at an early stage. ADMET properties are responsible for 60% of drug molecules failing during the final drug development process. An early prediction of these properties would dramatically reduce the costs of drug research. Further toxicity evaluation has been conducted on the compound that has favorable ADME properties to evaluate the harmful effects on humans and animals. Toxicology testing is an important element of drug development, but it is costly and time-consuming to conduct on animals [35]. The alternatives to *in silico* approaches have been adopted before drug development as they do not need animal trials and are also time and cost-effective. Following toxicity testing, we confirmed that the three compounds chosen were nontoxic or low-toxic.

MD simulation has emerged as a valuable tool in computer-aided drug design (CADD) as it allows for the examination of the dynamic behavior of compounds within a specific macromolecular environment. In this study, MD simulation was employed to assess the stability of the selected drug candidates when interacting with the target macromolecule. Key parameters such as RMSD, RMSF, and ligand-protein interactions were analyzed to evaluate the complex system. The results indicated optimal RMSD and RMSF values, along with favorable protein-ligand contacts, for all three compounds [36]. Based on the findings, the three selected compounds have demonstrated favorable characteristics and stability when interacting with the target protein. Consequently, these phytochemicals show promise as potential candidates for further development as antiviral drugs targeting the TiLV.

## Conclusions

The TiLV poses a substantial threat to the tilapia farming industry and to human food security, as it causes high mortality rates, and there is currently no effective antiviral medication available to combat the virus. This study aimed to identify potential natural antiviral drugs against the TiLV, with the goal of minimizing the frequency of virus infections. Three potential antiviral candidates, namely Procyanidins, delta7-Avenasterol, and Phytosterols, were selected through computer-aided drug design approaches such as homology modeling, molecular docking, ADMET, and MD simulation methods. These compounds hold promise in mitigating the economic losses in tilapia fisheries caused by the TiLV. This computational assessment will not only provide new information about the CRM1 insights into the TiLv but also will facilitate the treatment of fish virus in a significant and worthwhile manner. However, further *in vivo* and *in vitro* studies are recommended for the assessment of the effectiveness of these phytochemicals against TiLV in a practical way.

## Supporting information

**S1 Fig. Structural validation.** S1 Fig is included to show validation of the 3D structure of the CRM1 protein. (**A**) The Ramachandran plot statistics represent the most favorable, accepted, a disallowed region with a percentage of 97.984, 2.016, and 0.000%, respectively, and (**B**) the Z-score of refine gag protein −4.85.
(DOCX)

**S2 Fig. Compound/protein interactions.** Illustrations of the three selected compounds and their interactions and active poses with protein after re-docking.
(DOCX)

**S3 Fig. RMSDs of the sole ligands, phytosterols, avenasterol, procyanidin, and for comparison, leptomycin.** RMSDs are expressed in units of nanoseconds (ns) in this analysis. The ligand-hiMGAM snapshots of three phytochemical substances (Phytosterols, Avenasterol, and Procyanidin) and a marketable medication (Leptomycin B) were overlaid at the start and end timeframes. The ligands and bound hiMGAM proteins are visually distinguished by the colours green and red, respectively, in relation to the extracted frames at 0 ns and 100 ns.
(DOCX)

**S1 Table. CRM1 active sites and corresponding binding sites.** Calculated locations of CRM1 binding and interactions with each of the three selected proteins.
(DOCX)

**S2 Table. Docking scores of seventeen selected compounds.** These compounds were chosen for their relatively high binding affinities. The table illustrates the compounds, PubChem CID, their chemical names, molecular formulas, molecular weight, and docking scores of the seventeen selected compounds with highest binding affinities.
(DOCX)

**S3 Table. RMSD values through re-docking.** This table illustrates the three compounds selected for their binding affinities and their respective RMSD values analyzed through the process of re-docking.
(DOCX)

**S4 Table. RMS deviations.** This is a tabulation of RMS deviations as estimated using FlexX and SeeSAR software.
(DOCX)

**S5 Table. Stearic and electrostatic effects.** The steric and electrostatic effects arising from the spatial arrangement of three most preferable compounds are tabulated & presented here. (DOCX)

**S6 Table. MM/GBSA values.** Calculated MM/GBSA (binding free energy) and No Strain (NS) binding values and their variance calculated for each of the selected three compounds. (DOCX)

## Acknowledgments

The authors gratefully acknowledge the understanding and encouragement provided by the Ministry of Education and King Abdulaziz University, DSR, Jeddah, Saudi Arabia.

## Author Contributions

**Conceptualization:** Md Afsar Ahmed Sumon, Amer H. Asseri, Mohammad Habibur Rahman Molla, Mohammed Othman Aljahdali, Md. Rifat Hasan, M. Aminur Rahman, Md. Tawheed Hasan, Tofael Ahmed Sumon, Mohamed Hosny Gabr, Md. Shafiqul Islam, Burhan Fakhurji, Mohammed Moulay, Earl Larson.

**Formal analysis:** Mohammed Othman Aljahdali, Tofael Ahmed Sumon, Christopher L. Brown.

**Methodology:** Md Afsar Ahmed Sumon, Amer H. Asseri, Mohammad Habibur Rahman Molla, Mohammed Othman Aljahdali, Tofael Ahmed Sumon.

**Project administration:** Mohammad Habibur Rahman Molla, Christopher L. Brown.

**Supervision:** Md Afsar Ahmed Sumon, Amer H. Asseri, Mohammad Habibur Rahman Molla, Md. Tawheed Hasan.

**Writing – original draft:** Md Afsar Ahmed Sumon, Amer H. Asseri, Mohammad Habibur Rahman Molla, Mohammed Othman Aljahdali, Md. Tawheed Hasan, Tofael Ahmed Sumon, Christopher L. Brown.

**Writing – review & editing:** Md Afsar Ahmed Sumon, Amer H. Asseri, Mohammad Habibur Rahman Molla, Mohammed Othman Aljahdali, Md. Tawheed Hasan, Tofael Ahmed Sumon, Christopher L. Brown.

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
