## [Decision Letter · Decision Letter 0]

14 Aug 2023

PONE-D-23-18810Identification of Natural Antiviral Drug Candidates Against Tilapia Lake Virus: Computational Drug Design ApproachesPLOS ONE

Dear Dr. Brown,

Thank you for submitting your manuscript to PLOS ONE. After careful consideration, we feel that it has merit but does not fully meet PLOS ONE’s publication criteria as it currently stands. Therefore, we invite you to submit a revised version of the manuscript that addresses the points raised during the review process.

We look forward to receiving your revised manuscript.

Kind regards,

Abdulaziz Alouffi, PhD

Academic Editor

PLOS ONE

“As specified in the acknowledgments, This research was funded by Institutional Fund Projects under grant no. (IFPIP: 1349-130-1443). The authors gratefully acknowledge technical and financial support provided by the Ministry of Education and King Abdulaziz University, DSR, Jeddah, Saudi Arabia.  “

Reviewers' comments:

Reviewer's Responses to Questions

**Comments to the Author**

1. Is the manuscript technically sound, and do the data support the conclusions?

Reviewer #1: Yes

Reviewer #2: Yes

Reviewer #3: No

2. Has the statistical analysis been performed appropriately and rigorously? 

Reviewer #1: Yes

Reviewer #2: N/A

Reviewer #3: N/A

3. Have the authors made all data underlying the findings in their manuscript fully available?

Reviewer #1: Yes

Reviewer #2: No

Reviewer #3: Yes

4. Is the manuscript presented in an intelligible fashion and written in standard English?

Reviewer #1: Yes

Reviewer #2: Yes

Reviewer #3: Yes

5. Review Comments to the Author

Reviewer #1: 1. The English need improvement since there are some grammatical and syntax errors in the manuscript. For example,

• in line number 97, the words “a easy” may be as “an easy”;

• in line number 113, “in same” as “in the same”;

• in line number 137, “NCBI” as “the NCBI”;

• in line number 141, “anticipated” as “the anticipated”;

• in line number 145, “it in online” as “it online”;

• in line number 172, “had identified” as “identified”;

• in line number 173, “Lamarckian” as “the Lamarckian”;

• in line number 259, “2D” as “the 2D”;

• in line number 319, “has been” as “have been”;

• in line number 336, “compounds was” as “compounds were”;

• in line number 381, “simulations is” as “simulations are”;

• in line number 397, “lowest” as “the lowest”;

• in line number 404, “highest” as “the highest”.

The grammar mistakes which are not mentioned here are also to be checked and corrected properly.

2. There are some typing mistakes as well, and authors are advised to carefully proof-read the text. For example,

• in line number 118, the word “naturally” may be as “natural”;

• in line number 150, “refine” as “refined”;

• in line number 158, “plants” as “plant”;

• in line number 247, “Protein‐ Ligand” as “Protein‐Ligand”;

• Table 3, “Pharmakokinetics” as “Pharmacokinetics”;

• in line number 334, “shown” as “show”;

• in line number 340, “dynamic” as “dynamics”;

• in line number 398, “refine” as “refined”;

• in line number 421, “cost effective” as “cost-effective”.

The typos not mentioned here are also to be checked and corrected properly.

3. Check the abbreviations throughout the manuscript and introduce the abbreviation when the full word appears the first time in the abstract and the remaining for the text and then use only the abbreviation (For example, molecular dynamics (MD), CRM1, etc.,). Make a word abbreviated in the article that is repeated at least three times in the text, not all words to be abbreviated. The usage of abbreviations may be avoided in the keywords.

4. The author should uniformly follow some terms, for example, either “in silico” or “in- silico”.

5. The figure legends should be improved and a proper footnote should be given. All legends should have enough description for a reader to understand the figures without having to refer back to the main text of the manuscript. For example, the necessary abbreviations should be given.

6. The conclusion seems in general. All conclusions must be convincing statements on what was found to be novel, impact based on the strong support of the data/results/discussion. Moreover, the authors may also be included the limitation of the present findings for a better understanding of the manuscript.

Reviewer #2: Authors of the manuscript entitled “Identification of Natural Antiviral Drug Candidates Against Tilapia Lake Virus: Computational Drug Design Approaches” introduced computational approach for investigating the anti-viral potentiality for natural metabolites isolated from Heritiera fomes and Ceriops candolleana. Molecular docking analysis, in silico ADMET analysis, and molecular dynamics studies have highlighted the promising molecular reactivity and binding affinity of three identified hits. The manuscript is valuable in its field as it redeems publication following the address of these suggestions and comments:

1. The structure, nomenclature, and physiochemical properties of the identified 17 compounds should be presented even within the supplementary materials.

2. Section 3.2. Phytochemical and Protein Preparation, should be relocated to the materials and methods section rather than the in Results part.

3. Ramachandran plot should be presented at the supplementary materials.

4. The authors should adopt a positive control throughout the computational analysis to assess how the obtained results for the isolated natural metabolites would be of biological significance. Positive reference control could be a reported or market-approved inhibitor for the adopted CRM1 biotarget.

5. Validation of the docking protocol should be done through redocking protocol using the same docking algorism and parameters that are adopted for the investigated compounds has been recognized as a well-reported approach (See References; doi: 10.1021/jm0302997; doi: 10.1016/j.ejps.2020.105510; doi: 10.1016/j.bmcl.2019.02.031; doi: 10.1016/j.bioorg.2022.105770; doi: 10.3390/biology10050389). The ability of the molecular modelling simulation to replicate the ligand binding mode and residue-wise interaction patterns with low RMSD values (< 2.0 Å) between the native its redocked pose ensure the validity of the adopted protocol to provide binding pose of biological significance.

6. Authors provided comparative data regarding ligand binding modes through both highlighted polar hydrogen bonds and hydrophobic contacts. However, hydrogen binding should be presented within hydrogen bond distances as well as bond angles since hydrogen bond depend on both. Authors should mention the Hydrogen bond angles as well as their distances, since the strength of hydrogen bonding is based on both parameters in a way to ensure the adequacy of optimum hydrogen bonding.

7. Line 49, docking scores should be presented within their respective Kcal/mol units.

8. Through the RMSD and RMSF analysis, authors should illustrate trajectories for apo protein as well. This approach would better highlight the impact of compound’s binding on target through pinpointing flexible and immobile patterns for the protein ternary structures and amino acids in reference to the unliganded form. Difference RMSF (ΔRMSF = RMSFApo-Holo) could be adopted as well (please refer to doi: 10.1016/j.bmcl.2019.02.031 and doi: 10.1016/j.bmcl.2019.02.031).

9. Authors are advised to overlay of the initial, middle, and final frames (at 0 ns, 50 ns, and 100 ns, respectively) for each ligand-protein complex across the molecular dynamics simulations. This approach would provide great insights regarding the time-evolution orientation/conformation changes for both the protein and bounded ligands as well as the conserved and reformed ligand-amino acid bindings and close-range contacts.

10. Findings from the MM-GBSA free binding energy calculations as well as the constituting energy terms (VDW, electrostatic, solvation, SASA) should be represented in Figure or Table for better understanding the nature of ligand-target binding and the main driving energy terms for future structural improvements/optimization.

11. Authors should elaborate more on the discussion section through presenting comparative findings from reported literature studies that investigated other natural-isolated and/or even structural-related metabolites against the same target protein.

12. Finally, within the discussion sections, authors should highlight the takeaway messages that would be adopted in future lead optimization and development base on the docking, MD simulations, and ADMET studies. particularly, since two natural metabolites disobey Lipinski’s rule and cannot be considered as relevant clinical candidates. Prospective/recommended structure modifications to improve the ligand’s binding and interactions, as well as pharmacokinetics should be provided within the discussion and conclusion sections.

Reviewer #3: 1. Authors should discuss about the current clinical or preclinical or discovery status of CRM1 inhibitors as antiviral agents.

2. Is the selected target CRM1 validated clinically for viral inhibition??

3. Section 3.3- During the simulation procedure of molecular docking, the server-identified binding sites were used to build a receptor grid with grid box dimensions. Did the authors have generated grid box covering all the four pockets shown in Figure 1??

4. Section 3.4- top 20% of the 17 phytochemicals with the highest binding affinity were selected- How authors have come to this conclusion?? Is docking score alone sufficient to rank the compounds and that to on a predicted target protein and active pocket?? No comparison with standard or with binding interactions.

5. Have to perform MM-PBSA binding energy calculations to observe the energy variations and secondary structure analysis during the simulation time.

6. PLOS authors have the option to publish the peer review history of their article (what does this mean?). If published, this will include your full peer review and any attached files.

Reviewer #1: No

Reviewer #2: No

Reviewer #3: **Yes: **Manikanta Murahari

---

## [Author Response · Author response to Decision Letter 0]

7 Sep 2023

We have responded in detail to the reviewers responses and suggestions in our revised manuscript and in the response to reviewers document as uploaded.

---

## [Decision Letter · Decision Letter 1]

14 Sep 2023

PONE-D-23-18810R1Identification of Natural Antiviral Drug Candidates Against Tilapia Lake Virus: Computational Drug Design ApproachesPLOS ONE

Dear Dr. Brown,

Thank you for submitting your manuscript to PLOS ONE. After careful consideration, we feel that it has merit but does not fully meet PLOS ONE’s publication criteria as it currently stands. Therefore, we invite you to submit a revised version of the manuscript that addresses the points raised during the review process.

We look forward to receiving your revised manuscript.

Kind regards,

Abdulaziz Alouffi, PhD

Academic Editor

PLOS ONE

Reviewers' comments:

Reviewer's Responses to Questions

**Comments to the Author**

1. If the authors have adequately addressed your comments raised in a previous round of review and you feel that this manuscript is now acceptable for publication, you may indicate that here to bypass the “Comments to the Author” section, enter your conflict of interest statement in the “Confidential to Editor” section, and submit your "Accept" recommendation.

Reviewer #1: All comments have been addressed

Reviewer #2: (No Response)

2. Is the manuscript technically sound, and do the data support the conclusions?

Reviewer #1: Yes

Reviewer #2: Yes

3. Has the statistical analysis been performed appropriately and rigorously? 

Reviewer #1: Yes

Reviewer #2: Yes

4. Have the authors made all data underlying the findings in their manuscript fully available?

Reviewer #1: Yes

Reviewer #2: Yes

5. Is the manuscript presented in an intelligible fashion and written in standard English?

Reviewer #1: Yes

Reviewer #2: Yes

6. Review Comments to the Author

Reviewer #1: 1. There are some grammatical, alignments and typographical errors are noted in the manuscript and it should be thoroughly checked and corrected throughout the manuscript. For example,

• in line number 75, the words “to viral” may be as “to the viral”;

• in line number 79, “well characterized” as “well-characterized”;

• in line number 85, “disrupts” as “disrupt”;

• in line number 90, “tricomplex” as “tri-complex”;

• in line number 102, “investigation” as “the investigation”;

• in line number 103, “compounds” as “compound”;

• in line number 121, “best screened” as “best-screened”;

• in line number 140, “readily-available” as “readily available”;

• in line number 183, “BIOVA” as “the BIOVA”;

• in line number 208-209, and 331 “structure activity” as “structure-activity”;

• in line number 261, “compounds was” as “compounds were”;

• in line number 216, “single binding” as “single-binding”;

• in line number 273, “Tables” as “Table”;

• in line number 279, “Protein‐ Ligand” as “Protein‐Ligand”;

• “Pharmakokinetics” as “Pharmacokinetics”;

• in line number 319, “showed” as “shows”;

• in line number 387, “real life” as “real-life”;

• in line number 435, “protein– ligand” as “protein-ligand”;

• in line number 448, “era for” as “era of”;

• in line number 451, “simulations is” as “simulations are”;

• in line number 518, “computional” as “computational”.

2. This suggestion is not properly carried out The author should uniformly follow some terms either “in silico” or “in-silico”, for example, in line numbers 103 and 497 the authors have used “in silico” and in rest of the places “in-silico” has been used, it should be properly checked and corrected.

Reviewer #2: The authors addressed almost all suggestions and comments. However, two points should be addressed appropriately:

1. Adopting the protein's Cα as a reference for the analysed MD simulated trajectories is what meant by positive control for the in silico studies. owing to the lack of relevant experimental (wet lab) data, validation of the biological significance for the simulated ligands at docking and MD studies should be performed. Performing the same computational procedures for a literature reported or market-approved drug with experimental activity on the CRM1 biotarget and comparing them with those obtained for the isolated compounds would be relevant to extrapolate these computational data. Thus, it is advised to explore the docking and MD simulation of a literature reported or market-approved drug with CRM1 inhibition activity and compare these data wit those of the identified compounds.

2. The authors did not comprehend the concept of overlaying the 3D representation of the ligand-CRM1 complexes at initial, middle, and final frames. Trajectories at 0 ns, 50 ns, and 100 ns should be extracted for each ligand-CRM1 complex and then aligned. This would provide great insights regarding the time-evolution orientation/conformation changes for both the protein and bounded ligands as well as the conserved and reformed ligand-amino acid bindings and close-range contacts. authors are advised to refer to https://doi.org/10.3390/metabo13080942 for further guidance.

7. PLOS authors have the option to publish the peer review history of their article (what does this mean?). If published, this will include your full peer review and any attached files.

Reviewer #1: **Yes: **Prof. A. VIJAYA ANAND

Reviewer #2: **Yes: **Khaled M Darwish

---

## [Author Response · Author response to Decision Letter 1]

5 Oct 2023

All editorial recommendations have been accepted and impelemented in this revision, which is our second. That includes additional modeling work as suggested.

---

## [Decision Letter · Decision Letter 2]

16 Oct 2023

Identification of Natural Antiviral Drug Candidates Against Tilapia Lake Virus: Computational Drug Design Approaches

PONE-D-23-18810R2

Dear Dr. Brown,

We’re pleased to inform you that your manuscript has been judged scientifically suitable for publication and will be formally accepted for publication once it meets all outstanding technical requirements.

Kind regards,

Abdulaziz Alouffi, PhD

Academic Editor

PLOS ONE

Additional Editor Comments (optional):

Reviewers' comments:

Reviewer's Responses to Questions

**Comments to the Author**

1. If the authors have adequately addressed your comments raised in a previous round of review and you feel that this manuscript is now acceptable for publication, you may indicate that here to bypass the “Comments to the Author” section, enter your conflict of interest statement in the “Confidential to Editor” section, and submit your "Accept" recommendation.

Reviewer #1: All comments have been addressed

Reviewer #2: All comments have been addressed

2. Is the manuscript technically sound, and do the data support the conclusions?

Reviewer #1: Yes

Reviewer #2: Yes

3. Has the statistical analysis been performed appropriately and rigorously? 

Reviewer #1: N/A

Reviewer #2: Yes

4. Have the authors made all data underlying the findings in their manuscript fully available?

Reviewer #1: Yes

Reviewer #2: Yes

5. Is the manuscript presented in an intelligible fashion and written in standard English?

Reviewer #1: Yes

Reviewer #2: Yes

6. Review Comments to the Author

Reviewer #1: 1. There are some grammatical, alignments and typographical errors are noted in the manuscript and it should be thoroughly checked and corrected throughout the manuscript. For example,

• in line number 113, the words “natural” may be as “from natural”;

• in line number 209, “methodologies has” as “methodologies have”;

• in line number 245, “active” as “an active”;

• in line number 283, “Protein‐ Ligand” as “Protein‐Ligand”;

• in line number 302, “also various” as “various”;

• in line number 362, “others” as “other”.

2. This suggestion is not carried out properly. The usage of abbreviations may be avoided in the keywords, for example TiLV.

3. This suggestion is not carried out properly. The author should uniformly follow some terms, for example, either “in silico (in line number 103, 502)” or “in- silico (in line number 118, 140, 202 etc.,)”.

Reviewer #2: (No Response)

7. PLOS authors have the option to publish the peer review history of their article (what does this mean?). If published, this will include your full peer review and any attached files.

Reviewer #1: **Yes: **Prof. Dr. A. Vijaya Anand

Reviewer #2: **Yes: **Khaled M Darwish

---

## [Editor Report · Acceptance letter]

23 Oct 2023

PONE-D-23-18810R2 

Identification of Natural Antiviral Drug Candidates Against Tilapia Lake Virus: Computational Drug Design Approaches 

Dear Dr. Brown:

I'm pleased to inform you that your manuscript has been deemed suitable for publication in PLOS ONE. Congratulations! Your manuscript is now with our production department. 

Kind regards, 

on behalf of

Dr. Abdulaziz Alouffi 

Academic Editor

PLOS ONE